# Compare the performance of the models in art classification

**Wentao Zhao**[1,2], **Dalin Zhou**[3], **Xinguo Qiu**[1]*, **Wei Jiang**[1]

**1** College of Mechanical Engineering, Zhejiang University of Technology, Hangzhou, China, **2** School of Intelligent Transportation, Zhejiang Institute of Mechanical & Electrical Engineering, Hangzhou, China, **3** School of Computing, University of Portsmouth, Portsmouth, Uinted Kingdom

* xgqiu@zjut.edu.cn

**Data Availability Statement:** Painting-91 is avaliable at http://www.cat.uab.cat/~joost/painting91.html. WikiArt-WikiPaintings is avaliable at https://github.com/cs-chan/ArtGAN. MultitaskPainting100K is avaliable at http://www.ivl.disco.unimib.it/activities/paintings.

## Abstract

Because large numbers of artworks are preserved in museums and galleries, much work must be done to classify these works into genres, styles and artists. Recent technological advancements have enabled an increasing number of artworks to be digitized. Thus, it is necessary to teach computers to analyze (e.g., classify and annotate) art to assist people in performing such tasks. In this study, we tested 7 different models on 3 different datasets under the same experimental setup to compare their art classification performances when either using or not using transfer learning. The models were compared based on their abilities for classifying genres, styles and artists. Comparing the result with previous work shows that the model performance can be effectively improved by optimizing the model structure, and our results achieve state-of-the-art performance in all classification tasks with three datasets. In addition, we visualized the process of style and genre classification to help us understand the difficulties that computers have when tasked with classifying art. Finally, we used the trained models described above to perform similarity searches and obtained performance improvements.

## Introduction

Visual arts are omnipresent in our daily lives and appear in a variety of forms, such as home decorations and on public buildings (e.g., libraries, office buildings, and museums). People are attracted by the beauty and meaning in artworks; thus, there is a large demand for learning more details about artwork to improve people's lives.

With the development of computer science and computer technology, large numbers of visual artworks have been digitized. For example, the Google Art Project, which was launched by Google in cooperation with 17 international museums, allowed people worldwide to view panoramas of museum interiors and high-resolution images of artworks online rather than having to travel to the museums that house these works [1]. Art is largely created by people, and through experience and knowledge of specific features, people—especially art experts—can learn to easily identify various artists, painting styles and artwork genres [2]. However, to a computer, an image—whether a painting or a document image—contains only pixel values,

**Funding:** This work was supported in part by the Key Laboratory of E&M (Zhejiang University of Technology), Ministry of Education & Zhejiang Province(Grant No. EM 2016070101) The funders had no role in study design, data collection and analysis, decision to publish, or preparation of the manuscript.

**Competing interests:** The authors have declared that no competing interests exist.

which makes it difficult for computers to understand concepts such as beauty or emotions. Consequently, many studies have been conducted to investigate how to teach a computer to understand various painting characteristics. For classification tasks, the traditional approach is to first extract features identified by art and computer experts as being the most representative and then to use machine learning techniques for image categorization. For example, to categorize painting styles, the QArt-learn approach [3] was established based on qualitative color descriptors (QCD), color similarity (SimQCD), and quantitative global features. K-nearest neighbor (KNN) and support vector machine (SVM) models have been used to classify paintings in the Baroque, Impressionism and Post-Impressionism styles. References [4] applied KNN and SVM algorithms to achieve an emotional semantic understanding to classify a massive collection of images. These methods can help computers understand paintings in ways similar to experts; even nonexperts can understand how computers classify paintings.

Recently, an increasing number of papers have used convolutional neural network (CNN) models to solve classification problems. Many CNN models with different architectures have been proposed and achieve state-of-the-art performance on ImageNet. However, training CNNs requires datasets containing large numbers of labeled images to train the models and powerful computers. The recent successes of deep CNNs in solving computer vision tasks hinges on the availability of large hand-labeled datasets such as ImageNet [5], which contains more than 15 million hand-labeled high-resolution images representing approximately 22,000 different object categories. In the art classification field, the authors of [6] used CaffeNet [7], which is a slightly modified version of the AlexNet model [8], to evaluate the fine-tuning process using five different pretrained networks. Some of these innovations focus on the way data are imported and the various models used. In [9], three regions of interest (ROIs) were extracted from the input images using a regions of interest proposal module to identify the important areas of the image. To achieve this goal, this study used a deep multibranch neural network scheme. In [10], the input image was divided into five patches, and then a two-stage deep learning approach was used to classify the painting style. Even though deep CNNs can be used to classify paintings, they still need a set of labeled paintings for model training. Some unsupervised methods have also been applied to such problems. The authors of [11] used style labels to place artworks along a smooth temporal path. The training involved learning the style labels without any other features, such as the historical periods or contexts of the styles. Our goal is automatic classification and retrieval of paintings in different attributes by computer to help the viewer better understand the paintings. Furthermore, we explore how computers understand paintings.

The main contributions of this study are as follows.

- Many recently proposed models, such as ResNet and its variants (e.g., RegNet, ResNeXt, Res2Net, ResNeSt and EfficientNet), have achieved good performances on the ImageNet challenge and have been used to solve art classification problems. Thus, under the same conditions (transformation, optimization, and scheduling conditions), we test which of these models perform well on art classification tasks, including genre, style, and artist classification, and assess how transfer learning from models pretrained on general images affects the results. We count the best results for artist, style and genre classification on the three painting datasets using different models. Compared with the previous work, our results achieve state-of-the-art performance in all tasks with three datasets.

- We adopt three painting datasets to evaluate the classification accuracy of the models. These three datasets are Painting-91 [12], WikiArt-WikiPaintings [13], and MultitaskPainting100k [9], all of which are well-known benchmark datasets for art classification. We use both small and large datasets when testing the various tasks.

- By visualizing the embedding results by the models and constructing confusion matrices for the classification results, we investigate how computers perform classification and which styles are the most difficult for computers to classify. Finally, we build a painting retrieval system based on the trained classification models and find that it achieves good performance.

In this paper, we first introduce the datasets and classification tasks in the Datasets and Classification Tasks section. We introduce the CNNs we used to compare the performance in art classification in the Convolutional Neural Network Models section. The experimental setup including the transfer learning and training settings are presented in the Experimental Setup section. We present the results and provide some visualizations to further explain how computers interpret paintings. Additionally, we show that features derived from networks can be employed to retrieve image similarity in terms of artist, genre and style in the Results and Discussion section. Finally, we draw a conclusion and propose future work in the Conclusion section.

## Datasets and classification tasks

With the goal of classifying paintings, we use three different painting datasets to perform the classification tasks as described below.

### Painting-91 dataset

This dataset includes 4,266 painting images painted by 91 different artists. Of these paintings, we use 2,275 paintings for training and 1,991 for evaluation in the artist classification tasks. In addition, the Painting-91 dataset contains 2,338 paintings which are classified into 13 different styles. Of these, we used 1,250 for training and the other 1,088 for evaluation. Because the painting styles are distributed uniformly among these paintings and the dataset is not particularly large, it is convenient for training and has been widely used in art classification tasks.

### WikiArt-WikiPaintings dataset

The images in the WikiArt dataset were obtained from wikiart(http://www.wikiart.org). This dataset is still being updated; thus, to ensure a fair comparison with other models, we adopted the same dataset used in [14] for this study. The WikiArt painting dataset [15] is a well-organized collection that consists of more than 80,000 high-art images. This dataset includes more than 1,000 artists, 27 different styles and 45 different genres. Due to the limited number of samples available in some classes, we ignored some images in the classification tasks. Finally, for 23 artists with more than 500 paintings, we selected paintings from each category for the artist classification task. For 10 genres represented by more than 1500 paintings, we selected paintings from each category for the genre classification task. For the style task, we used all paintings for classification; each style is represented by at least 90 images.

### MultitaskPainting100K dataset

The dataset used in our work stems primarily from the Kaggle competition. Most images come from WikiArt.org; the remainder are paintings created by artists specifically for the competition. The original goal on this dataset was to predict whether a pair of images were painted by the same artist. To accomplish the artist, style and genre classification tasks, some images were discarded to ensure that every class had at least 10 images. Subsequently, the number of images was reduced from 103,250 to 99,816, forming a total of 1,508 artists, 125 styles and 41 genres. The dataset is randomly split into 2 parts: 70% in the training set and 30% in the testing

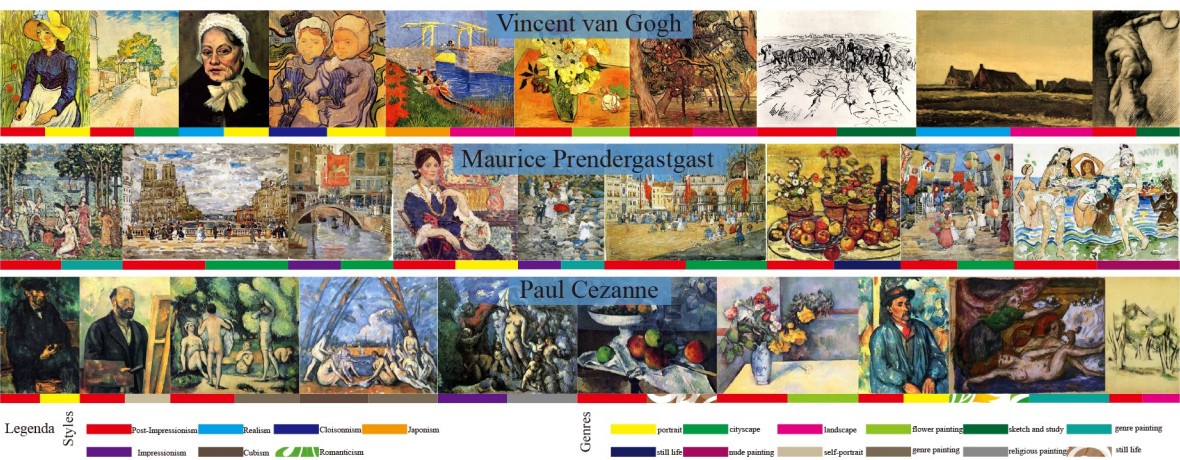

**Fig 1. Paintings from the MultitaskPainting100k dataset.** Each row shows samples from a different artist. We included different styles and genres based on the color coding. We state that all the paintings are all in the public domain(courtesy wikiart.org).

set. Fig 1 shows some examples. Each row contains samples from a different artist. Among the artists are Vincent van Gogh, Maurice Prendergast, and Paul Cezanne. The different colors used are shown under the pictures; these represent the different genres and styles. Some artists paint in multiple genres and styles. Different from the WikiArt dataset, there are nearly 100,000 images in the MultitaskPainting100K dataset. Each image record includes 3 attributes, allowing every image to be used for artist, style and genre classification tasks. The main goal in this study is to recognize and classify artists, genres and painting styles. Any given artist may produce works in multiple styles and genres, as shown in Fig 1; these examples are indicative of how difficult it is for a computer to classify the paintings.

These 3 datasets represent different priorities. Compared to the others, Painting-91 is a small dataset. Thus, models can be trained on this dataset quickly, and we can study how to best perform author, genre, and style classification on small datasets. We filtered the WikiArt-WikiPaintings dataset to ensure that sufficient samples are available in all classes for the tasks. Even after filtering, this dataset still contains 20,000, 65,000 and 81,000 images in the artist, genre and style classification tasks, respectively. The MultitaskPainting100K dataset has nearly 100,000 images available for each task. In total, there are 1,508 artists, 125 styles and 41 genres. This means that some categories have only a small number of images and that the training data are unbalanced, which makes the classification tasks more difficult than for balanced training data. Total number of images and classes per task are introduced in Table 1. The state-of-the-art results in previous works and ours will be introduced in the Results and Discussion section.

## Convolutional neural network models

AlexNet [8] won the championship in the ImageNet Large Scale Visual Recognition Challenge on September 30, 2012. An increasing number of state-of-the-art deep CNN models trained on very large datasets such as ImageNet have been made available for research purposes. Models such as VGG16 [21], ResNet50 [22], and ResNet152 have been proposed and used in many scenes. The compared deep CNNs were typically developed under a fixed resource budget. Improving their accuracy also tends to make their architectures more complex. It has been

**Table 1. State-of-the-art results for artist, style and genre categorization, including samples and classes in each tasks.**

| | | Artist | | | Style | | | Genre | | |
|---|---|---|---|---|---|---|---|---|---|---|
| dataset | references | sample | classes | Acc.(%) | samples | classes | Acc.(%) | samples | classes | Acc.(%) |
| Painting-91 | [16] | 4,266 | 91 | 64.32 | 2,338 | 13 | 78.27 | | | |
| | ours | | | **71.27** | | | **79.23** | | | |
| WikiArt | [14] | 19,050 | 23 | 76.11 | 81,444 | 27 | 54.5 | 64,993 | 10 | 74.14 |
| | [15] | 18,599 | 23 | 63.06 | 78,449 | 27 | 45.97 | 63,691 | 10 | 60.28 |
| | [17] | 17,100 | 57 | 77.7 | | | | | | |
| | [18] | | | | | | | 79,434 | 26 | 61.15 |
| | [6] | 20,320 | 23 | 81.94 | 96,014 | 27 | 56.43 | 86,087 | 10 | 77.6 |
| | [19] | | | | 26,400 | 22 | 66.71 | | | |
| | [20] | 9,766 | 19 | 88.38 | 30,825 | 25 | 58.99 | 28,760 | 10 | 76.27 |
| | ours | 19,050 | 23 | **91.73** | 81,444 | 27 | **69.97** | 64,993 | 10 | **78.03** |
| Multitask Painting100k | [9] | 99,816 | 1508 | 56.5 | 99,816 | 125 | 57.2 | 99,816 | 41 | 63.6 |
| | ours | | | **65.50** | | | **63.15** | | | **67.83** |

found that models with deeper network depths, widths and resolutions achieve better accuracy. ResNet has been extended from ResNet-18 to ResNet-200 by increasing the network depth. However, deeper networks also have greater computational complexity, and the number of parameters increases dramatically.

The main deep CNN architecture used in our experiment is ResNet and its variants; these include Res2Net, ResNeXt, RegNet and ResNeSt and EfficientNet. ResNet introduced a shortcut connection to solve the degradation problem. Fig 2A shows the bottleneck block of Resnet50. The three layers are $1 \times 1$, $3 \times 3$ and $1 \times 1$ convolutions, where $1 \times 1$ layers are responsible for reducing and then restoring dimensions, making the $3 \times 3$ layer a bottleneck with smaller input/output dimensions. Res2Net [23] constructed hierarchical residual-like connections within a single residual block to increase the number of scales that the output features can represent (Fig 2B). The model represents multi-scale features at a fine-grained level and increases the receptive domain range of each network layer whenever it passes a $3 \times 3$ filter. ResNeXt [24] is constructed by repeating a building block that aggregates a set of transformations with the same topology. This strategy uses "cardinality" (the size of the set of transformations) to create different paths in blocks, which are subsequently merged (e.g., Fig 2C, a block of ResNeXt with cardinality = 32, with roughly the same complexity as a block

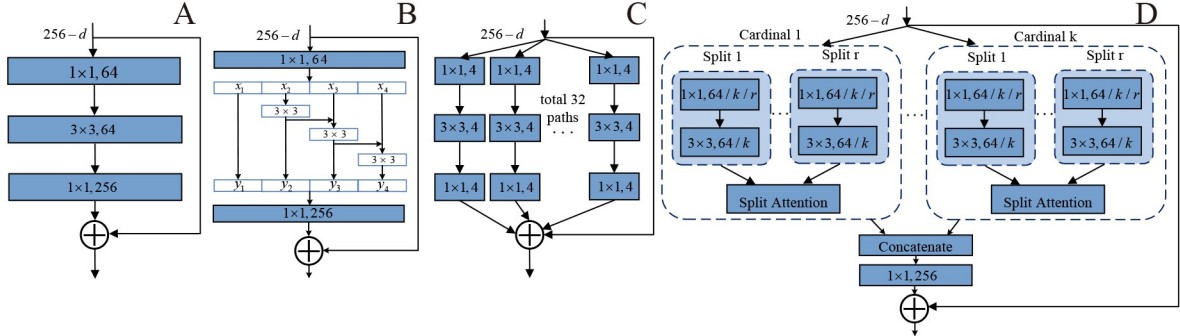

**Fig 2. Comparing the block of ResNet, Res2Net, ResNeXt and ResNest.** A. ResNet block. B. Res2Net block. C. ResNeXt block. D. ResNest block.

of ResNet). The different paths occur on behalf of different representation subspaces. This causes the models to learn different information from different representation subspaces. The experiment on the ImageNet-1K dataset shows that increasing cardinality is more effective than going deeper or wider when increasing the capacity. ResNeSt [25], also called a split-attention network, introduced the concept of the split-attention module that enables attention across feature-map groups. Fig 2D depicts an overview of a split-attention block, which is a computational unit consisting of feature-map groups and split attention operations. Like a ResNeXt block, the features are first divided into cardinal groups using cardinality hyperpara-meter $K$. Then, the cardinal groups are divided into $R$ splits. Experiments show that the ResNeSt models outperform other networks with similar model complexities and can be well used for downstream tasks. RegNet [26] design network design spaces that parametrize popu-lations of networks. The author explored the structure aspect of network design and arrived at a low-dimensional design space consisting of simple, regular networks called RegNet. The researchers first created an initial design space [24] called AnyNet that contains a stem, body and head. Most experiments use the standard residual bottlenecks block called X block and the AnyNet design space built on it as AnyNetX. After designing some rules and searching for the best models, the final designed model was called RegNetX. RegNetY was located by evaluating RegNetX using the Squeeze and Excitation (SE) operation [27]. The core point of RegNet is very simple: widths and depths of good networks can be explained by a quantized linear func-tion. EfficientNet [28] systematically studies model scaling and identifies that carefully balanc-ing network depth, width, and resolution can lead to better performance. The author used a neural architecture search to design a new baseline network and scale it up to obtain a family of models. In detail, the author proposed a new compound scaling method as shown in (1):

$$\begin{cases} \text{depth} : d & = \alpha^{\phi} \\ \text{width} : w & = \beta^{\phi} \\ \text{resolution} : r & = \gamma^{\phi} \end{cases} \tag{1}$$

where s.t.$\alpha \cdot \beta^2 \cdot \gamma^2 \approx 2$ and $\alpha \geq 1, \beta \geq 1, \gamma \geq 1$. $\phi$ is a user-specified coefficient that controls model scaling, while $\alpha, \beta, \gamma$ can be determined by a small search and specify how to assign these extra resources to network width, depth, and resolution, respectively. Because the model scaling does not change the layer in the baseline network, the authors search spaces to produce an efficient network, named EfficientNet-B0. By scaling up the baseline network with different $\phi$ using Eq (1), the authors obtained EfficientNet-B1 to B7.

## Experimental setup

### Transfer learning

Transfer learning is a cornerstone of computer vision, and many different image-related classi-fication tasks [29, 30], which work with datasets of limited size, achieve state-of-the-art perfor-mance by using transfer learning. Previous works have proved that it achieves great performance by using a pretraining model and fine-tuning it [31–35]. To adapt the models for art classification, we modified the last fully connected layer to match the number of classes in each task. We use random initialization in last fully connected layer and weights pretrained in ImageNet in the other layers.

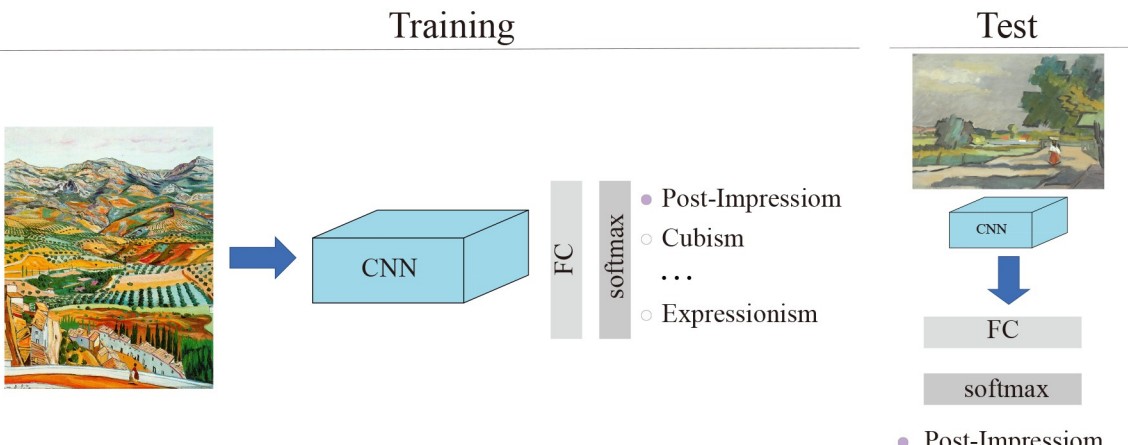

**Fig 3. The overall general architecture of the model.** We modified the last full connection layer of CNN to meet the different categories of paintings in different tasks. There, we show the style categories. We state that all the paintings are all in the public domain (courtesy wikiart.org).

## Training settings

In our experiments, we designed a unified framework that can learn representations containing the artist, genre and style of visual arts from a large number of digital artworks with multiple labels separately. Then, the feature extracted from the framework can be used to classify the paintings. Furthermore, it can be used to retrieve similar paintings, as introduced in the Results and Discussion section. To extract features and use them in classification tasks, we built an overall general architecture defining the entire system as shown in Fig 3. The total system contains 2 sections. In the training section, the visual embeddings representing the visual appearance of paintings are extracted. The weights of the CNN can be learned via appropriate learning methods. In the test section, the CNN model trained to extract the visual embedding is used to classify paintings based on the extracted painting features.

The pseudocode for training a neural network with a mini-batch and calculating the classification rate using test paintings in our experiment is shown in Algorithm 1. In each iteration, we sample $b$ images to compute the gradients and then update the network parameters. We evaluate the model performance in each epoch and use the maximum accuracy as the final result after $K$ epochs of training. All networks were fine-tuned using stochastic gradient descent (SGD) [36] with L2 regularization, which skips bias and batch normalization, a momentum of 0.9, and a weight decay of 0.0001. In all experiments, the following data augmentation techniques were applied to expand the small amount of training data:

1. RandomResizedCrop. A random crop operation with range of size between 0.08 and 1.0 of the original size and aspect ratio (from 3/4 to 4/3) of the original aspect ratio were conducted. The cropped image was finally resized to 224*224 using random interpolation.

2. RandomHorizontalFlip. The given image was randomly flipped horizontally at a preset probability of 50%.

3. The Python import library (PIL) image was converted to a tensor and normalized using the mean and standard deviation.

**Algorithm 1**: Train a neural network and get the accuracy for painting classifiation
```
Input: initialize(net)
```

```
   Output: Test Acc.
1  for epoch = 1,...,K do
2    for batch = 1,2,...,#images/b do
3      images ← uniformly random sample b images;
4      x, y ← progress(images);
5      z ← forward(net, x);
6      ℓ ← loss(z, y);
7      grad ← backward(ℓ);
8      update(net, grad);
9      Acc = Eval(net,test_k);
10   end
11 end
12 Acc = Max(Acc)
```

For the test data, we first resized the image to 256 * 256 and cropped the images at the center to 224 * 224 (except EfficientNet-B3, which we first resized to 331 * 331 and then cropped the images at the center to 300*300). Then, we converted the PIL image to a tensor and normalized it using the mean and standard deviation. For the learning rate, when training on Painting-91, we chose 0.1 as the initial learning rate for a batch size of 64. When the batch size changed to a larger batch size ($b$), we increased the initial learning rate to 0.1 * $b$/64. For example, for the larger datasets WikiArt and MultitaskPainting100K, to improve the training speed without sacrificing model accuracy, we increased the batch size to 128, which means that the initial learning rate was set to 0.2. To save time and memory, we used mixed precision training in this experiment. We trained the model using a Titan RTX, which boasts 16.3 Tflops of single-precision and 130 Tflops of tensor performance (FP16 w/FP32 Acc). When using FP16 to calculate the weights, one major problem is that the parameter values in the models may be out of range because the dynamic range of FP16 is narrower than that of FP32, which interferes with the training process. To solve these problems, we followed [37], which suggested initially storing all the weights, activation and gradients in FP16 and copying the weights to FP32. Then, FP16 was used to calculate the gradients, and FP32 was used to update the weights. To adjust the learning rate, we started with a learning rate warmup period that lasted for the first several epochs and used cosine learning rate decay. The learning rate $\eta_t$ was computed as shown in (2):

$$
\begin{cases}
\eta_t = \dfrac{t}{t_0}\eta & t \leqslant t_0 \\[2ex]
\eta_t = \dfrac{1}{2}\left(\eta - \eta_{cool}\right)\left(1 + \cos\dfrac{(t - t_0)\pi}{T}\right) & t_0 < t < T \\[2ex]
\eta_t = \eta_{cool} & t > T.
\end{cases}
\tag{2}
$$

where $\eta$ is the initial learning rate, $t_0$ is the number of warmup epochs, and $T$ is the total number of epochs. Our experiments used 160 epochs. This type of scheduling is called "cosine decay" [38]. After the cosine decay, we trained the models for several epochs (10 in our experiments) using the learning rate $\eta_{cool}$. Therefore, there are 170 ($K$) total epochs in our experiments. The classifier is a fully connected layer, and its output is used to compute a classification loss using a cross-entropy loss function. This loss can be calculated as shown in (3):

$$
\begin{aligned}
\text{loss}(x, class) &= -\log\left(\frac{\exp\left(x[class]\right)}{\sum_j \exp\left(x[j]\right)}\right) \\[2ex]
&= -x[class] + \log\left(\sum_j \exp\left(x[j]\right)\right).
\end{aligned}
\tag{3}
$$

**Table 2. The results of the classification in Painting-91.**

| models | ResNet | Res2Net | ResNeXt | RegNetX | RegNetY | ResNeSt | EfficientNet |
|---|---|---|---|---|---|---|---|
| Artists/pre | 58.92 | 58.81 | 59.37 | 65.44 | 66.60 | 60.07 | **71.27** |
| Styles/pre | 66.54 | 70.13 | 67.37 | 73.35 | *59.31 ± 7.80* | 62.96 | **79.23** |
| Param_count | 25.56 | 25.06 | 25.03 | 26.21 | 30.58 | 27.48 | 12.23 |
| ImageNet | 79.04 | 78.15 | 79.77 | 79.07 | 79.72 | 80.97 | 82.08 |
| Artists | 33.95 | 35.86 | 36.06 | 35.06 | 35.01 | 35.01 | **42.84** |
| Styles | 51.01 | **53.40** | 51.47 | 48.44 | 49.72 | 48.16 | 45.50 |

where *x* and *class* are the output of the classifier and the targeted label of the painting being learned, respectively. We ran our experiments using Pytorch 1.5.0 [39] in Ubuntu 18.04 with Titan RTX and Intel i9 10900k. All the pretrained models we used are from PyTorch Image Models [40](timm) library.

## Results and discussion

### Results

In total, we tested 7 models with different architectures in the experiments: ResNet50, Res2Net50_14w_8s, ResNeXt50_32*4d, RegNetX_064, RegNetY_064, ResNeSt50 and efficientnet_B3. For better representation, the models are abbreviated in Tables 2–4. The model Res2Net50_14w_8s means we separated the block into 8 subsets after a 1*1 convolution, each with 14 channels. For ResNeXt50_32*4d, the number 32 represents the cardinality, and 4d means that there are 4 channels in each block group (bottleneck width = 4d). In RegNetX_064 and RegNetY_064, the number 064 denotes Gflops. For ResNeSt50, we make cardinality as 1 and radix as 2 with base_width 64. To ensure a fair comparison, the numbers of parameters required by these models did not differ widely (except EfficientNet_B3, which requires a large amount of CUDA memory), and we largely used the same hyperparameters for the models in the experiments; the details are shown in the Experimental Setup section. The models were either pretrained with weights from ImageNet or not pretrained. In Tables 2–4, the first 2 rows use the models pretrained on ImageNet, and transfer learning is applied before the art classification tasks. The third row shows the parameter count for each model. The fourth row shows the model image classification rate training on ImageNet, and the last 2 rows show the results after training the models starting with random weights. The numbers in bold print show the highest accuracy values from the art classification tasks. The numbers in italic represent

**Table 3. The results of the classification in Wikiart.**

| models | ResNet | Res2Net | ResNeXt | RegNetX | RegNetY | ResNeSt | EfficientNet |
|---|---|---|---|---|---|---|---|
| Artists/pre | 88.03 | 88.14 | 88.29 | 89.08 | *76.29 ± 1.50* | 88.17 | **91.73** |
| Styles/pre | 65.76 | 65.97 | 66.62 | 67.10 | 69.51 | **69.97** | 69.19 |
| Genres/pre | 76.69 | 76.54 | 76.74 | 76.63 | 77.41 | 77.94 | **78.03** |
| Param_count | 25.56 | 25.06 | 25.03 | 26.21 | 30.58 | 27.48 | 12.23 |
| ImageNet | 79.04 | 78.15 | 79.77 | 79.07 | 79.72 | 80.97 | 82.08 |
| Artists | 79.72 | 80.07 | 80.97 | 81.98 | 82.09 | **83.84** | 82.39 |
| Styles | 62.24 | 63.32 | 63.25 | 63.11 | 65.40 | **66.81** | 64.91 |
| Genres | 75.77 | 75.94 | 75.90 | 75.38 | 76.12 | **77.07** | 75.66 |

Table 4. The results of the classification in MultitaskPainting100k.

| models | ResNet | Res2Net | ResNeXt | RegNetX | RegNetY | ResNeSt | EfficientNet |
|--------|--------|---------|---------|---------|---------|---------|--------------|
| Artists/pre | 59.05 | 61.28 | 59.93 | 59.97 | 61.22 | 62.78 | **65.50** |
| Styles/pre | 59.41 | 60.81 | 60.73 | 60.60 | 62.80 | 62.64 | **63.15** |
| Genres/pre | 65.77 | 66.31 | 66.27 | 65.99 | 66.82 | **67.83** | 66.99 |
| Param_count | 25.56 | 25.06 | 25.03 | 26.21 | 30.58 | 27.48 | 12.23 |
| ImageNet | 79.04 | 78.15 | 79.77 | 79.07 | 79.72 | 80.97 | 82.08 |
| Artists | 52.46 | 53.34 | 53.99 | 54.19 | 53.98 | 57.43 | **60.93** |
| Styles | 56.16 | 56.99 | 56.97 | 56.69 | 59.01 | **60.94** | 58.64 |
| Genres | 64.37 | 65.23 | 64.45 | 64.76 | 65.90 | **66.47** | 65.57 |

mean ± variance. During the experiments, for some accuracies using RegNetY_064, we found that the training results are unstable. Therefore, the results were calculated 4 times, and the results are presented as the mean ± variance. As listed in Table 2, for the small datasets, using models pretrained with weights on ImageNet is more useful than training with random weights. This result indicates that art classification is related to real-world images. The weights pretrained on ImageNet are also useful in art classification. We can see that the variant version of ResNet50 performs better than ResNet50 itself. EfficientNet_B3 achieved 71.27% and 78.22% on artist and style classification, respectively. Table 3 shows that as the training datasets become larger, the power of the various structures emerges. ResNeSt50, which performs well when pretrained on ImageNet, also performs well on the art classification in both pretrained and unpretrained versions. Among the 6 result values, 4 of them are better than those of the other models. This phenomenon shows that the model architectures excellent in ImageNet also perform well on the task of art classification, which has strong applicability. Table 4 also shows this phenomenon. Categories with only a few images were not filtered from the MultitaskPainting100k dataset; thus, it still contained some categories with few images. This led to classification accuracy problems; the models performed worse on MultitaskPainting100k than on the WikiArt dataset.

The average accuracy for the tasks of artist, style and genre classification in previous work is reported in Table 1. [16] use learned deep correlation features (LDCF) from Gram to achieve success rates of 64.32% and 78.27% in the artist and style tasks in Painting-91, respectively. [14] achieved the best results with fine-tuning an Alexnet network pretrained on the ImageNet dataset. [15] explored how different features and metric learning approaches influence the classification results and achieved the best results with the feature fusion method. [18] used the Sun database [41] to augment some WikiArt classes and used ResNet34, which was not initialized on ImageNet, for classification in genre tasks. [17] used ResNet-18 with pretrained weights from the ImageNet dataset in artist tasks. [6] used a fine-tuned CaffeNet pretrained on domain-specific datasets to achieve the best performance. [19] introduced a two-stage image classification approach including a deep convolutional neural network (DCNN) and a shallow neural network to improve the style classification accuracy. [20] used the RGB and brush stroke information to classify fine-art painting images. [9] proposed the MultitaskPainting100k dataset and used a spatial transformer network (STN) that was introduced by [42] with the injection of HOG features to achieve 56.5%, 63.6% and 57.2% success rates in artist, genre and style tasks, respectively. The WikiArt dataset is from the WikiArt website, and as the number of paintings on websites increases over time, there are differences in the selection methods of paintings for different studies, so the number of paintings and the number of categories differ among the algorithms. We also report the number of classes and samples

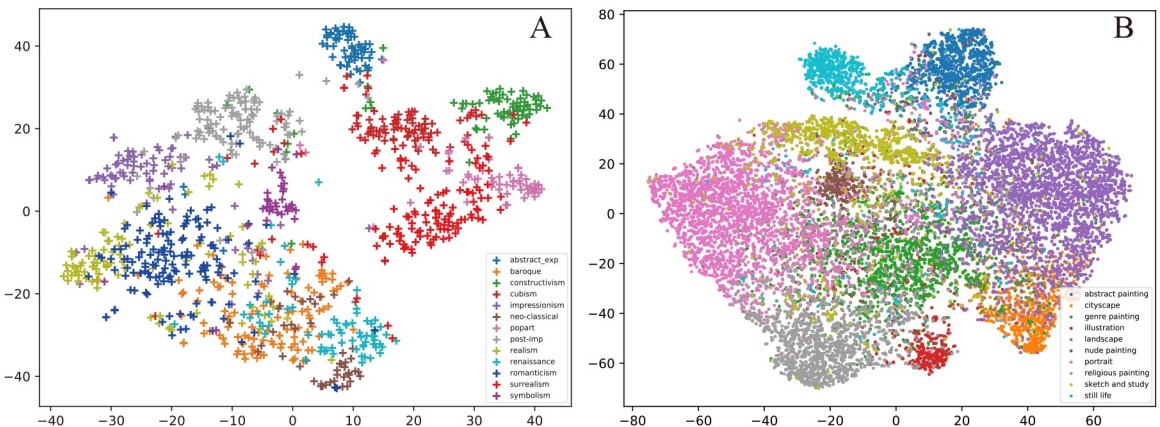

**Fig 4. Embedding of the paintings projected in Painting-91 and WikiArt using t-SNE.** Each node is a painting, and the coloring is mapped to the style attribute and genre attribute. A. Painting-91 with style attribute. B. WikiArt with genre attribute.

considered in the experiments presented in each paper. Although the dataset configurations are not identical, because the data are from the same sources, the results still have certain reference value. By comparing the previous work using different tricks, we can find that our results which simply optimizing the model structure, achieve state-of-the-art performance on all tasks in the 3 datasets.

## Visualization and discussion

We used different models to encode the paintings in Painting-91 and WikiArt. Then, the t-SNE [43] algorithm was used for dimensional reduction as shown in Fig 4. We observe that models can learn more visual embeddings and that the paintings with the same label are close to each other. The confusion matrices [44] of style and genre classification tasks in Painting-91 and WikiArt are shown in Fig 5. In each picture, the Y axis represents the true label, while the X axis represents the predicted label. It can be found that by modifying the model architecture, the recognition rate of each category can be improved effectively. Among all points, a few observations are worth attention. By using different models, the recognition rate is effectively improved, but the recognition distributions of different types are roughly the same. The true label in Fig 5 corresponds to the different colors in Fig 4. Because the embedding results are used to predict the label, the predicted label is closely associated with the location of nodes in Fig 4. Many details can be found from style embedding in Painting-91; the points that have labels such as abstract expressionist, cubism, popart, surrealism and symbolism are clustered together, which makes these categories well distinguished. In contrast, the various aspects of styles such as Realism, Romanticism, Renaissance, Neoclassicism and Impressionism intersect with each other and seem to be difficult for a computer to classify. For example, the model often confuses other painting styles with those of Baroque because the Baroque points in Fig 4A are mixed up among the other points. Thus, the model can not classify this style unambiguously. The model often misclassifies Baroque paintings as Neoclassical and Romanticism paintings because the Euclidean distance in Fig 4A is too close. By further observation in Fig 5B, we find that the most obvious type of style is abstract expressionism (95%), which was developed in New York in the 1940s and is often characterized by gestural brush-strokes or mark-making, giving the impression of spontaneity. The relationship between Renaissance, Baroque and Neoclassical makes it difficult for the computer to distinguish them. Realism in

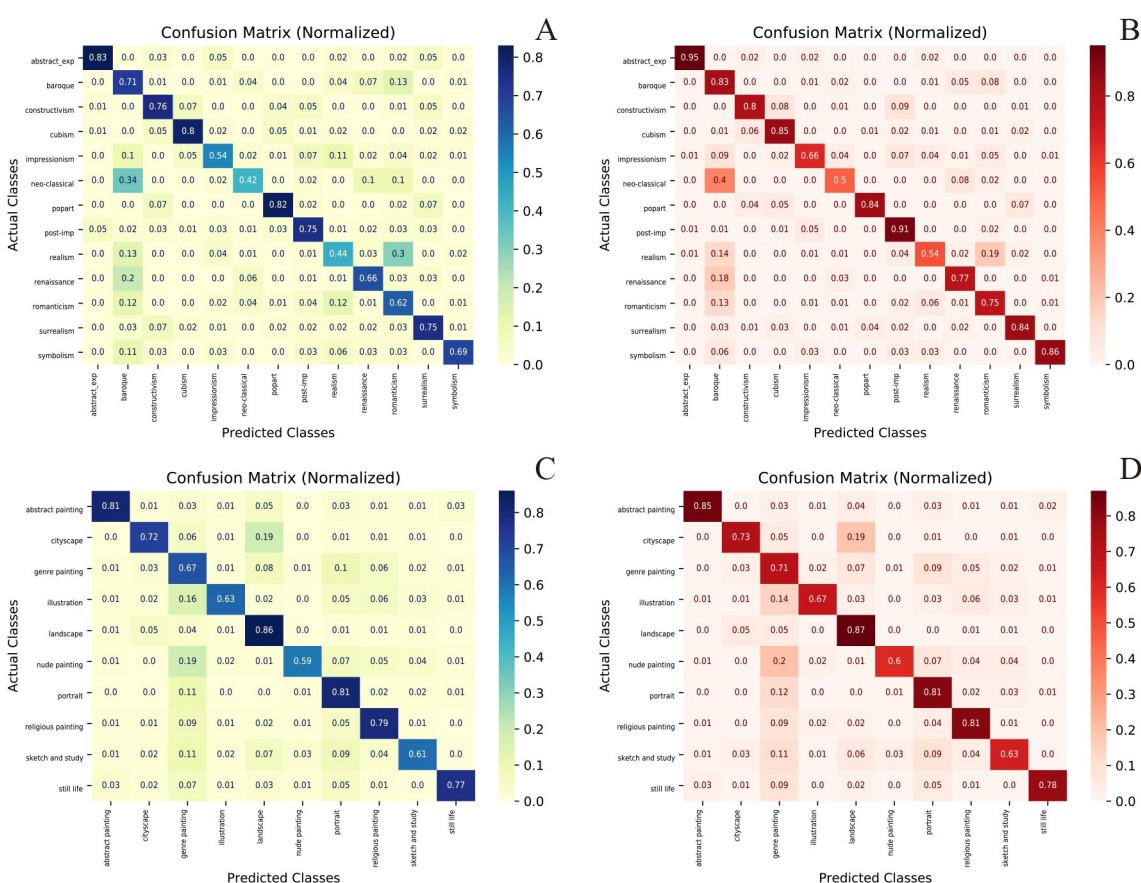

**Fig 5. Confusion matrix of the different classification tasks.** A. style classifications in Painting-91 using Resnet50. B. style classifications in Painting-91 using EfficientNet. C. genre classifications in WikiArt using Resnet50. D. genre classifications in WikiArt using EfficientNet.

the arts is generally the attempt to represent subject matter faithfully. The Realist painters rejected Romanticism, which had come to dominate French literature and art, with roots in the late 18th century, but the computer often mistaken Realism for Romanticism due to the continuity of time between the two styles. In genre classification on WikiArt, paintings belong to abstract painting, landscape and portrait are clustered (Fig 4B), which makes these categories well distinguished. Paintings achieve 85%, 87% and 81% recognition rates in each category when using EfficientNet-B3, respectively (Fig 5D). The paintings belonging to nude painting are mixed up with other paintings, and there are only 60% accuracy rate for these types of paintings. By further observation in Fig 5D, a cityscape is mistaken for a landscape because they all include outdoor scenes; illustration and nude painting are often referred to as genre painting because they all contain people.

## Similarity search

Many content-based search engines exist and have been applied to many areas. Such searches are similar to Google Images, Baidu content searches, or Taobao (for commodity recommendation). In the following, we present some interesting results obtained when the proposed method is used for similarity search. Given one painting as an input, we extract 3 sets of

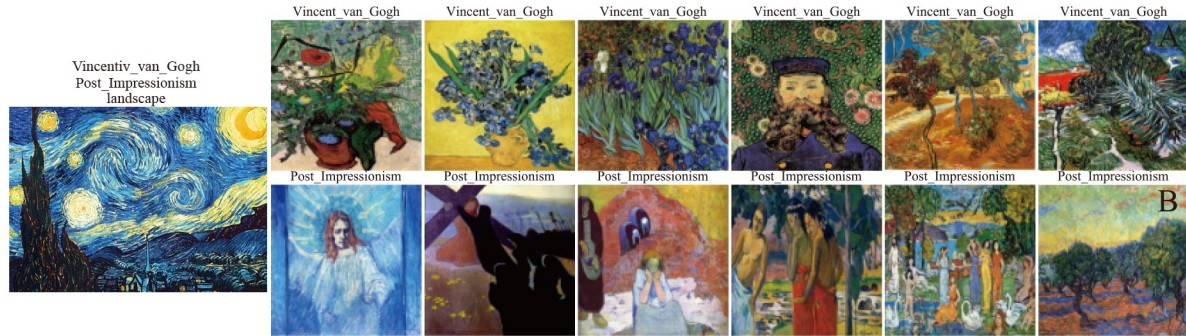

**Fig 6. Similarity search for the painting Starry Night, by Vincent Van Gogh.** A. The similarity results show the top six paintings retrieved using artist features. B. The similarity results show the top six paintings retrieved using style features. We state that all the paintings are all in the public domain(courtesy wikiart.org).

features using different models. Each set of features can then be used to compute a similarity score in terms of artists, styles and genres with the search library, which contains painting embedding from WikiArt. The cosine and Euclidean distances are typically used for similarity measurements. Here, we use the cosine distance to calculate the similarity between different paintings. The image retrieval database we used is WikiArt.

Starry Night(Fig 6) is an oil-on-canvas painted by the Dutch Post-Impressionist painter Vincent van Gogh. We use the models trained to classify artists and styles to retrieve the paintings. We can find that the most similar paintings have the same attributes. In addition, when we use the genres to retrieve the paintings, the most similar paintings have the same attributes, too. Another search object for paintings is called the key. This painting is one of Jackson Pollock's Accabonac Creek series, which marks a crucial moment in Pollock's evolution as an artist. This quasi-surrealist painting was created on the floor of an upstairs bedroom, and the artist worked on it directly from all sides. We extracted the painting from Painting-91, and the system predict the painting was created by Pablo_Picasso because the artist is not listed in the label on the WikiArt dataset. This makes it difficult to retrieve the author, because abstract expressionism completely rejects all forms (biomorphic as well as geometric) and uses color as the only tool for expression. When referring to style and genre retrieval, all the retrieval results are in the same category as the retrieved painting. From these 2 examples, we can find that the retrieval system we have built is useful for painting retrieval.

## Conclusion

In this study, we used CNNs to perform various types of artwork classifications. Approximately 7 models that work well on general image classifications were trained, applied to the classification tasks and performed well. These results indicate that both the model architectures and the parameter weights pretrained on generalized image datasets are important. Starting with pretrained parameters has a greater impact on small datasets than on larger datasets. In addition, models that perform better on ImageNet perform better on the artwork classification tasks. Thus, it would appear that the art world has similarities to the real world and that model enhancements that improve real-world classification rates can also be used in the art world. In addition, we used the models trained to classify styles and genres in different datasets to analyze how computers perform art classification. Finally, we built a painting retrieval

similarity search engine using the trained classification models and used it to retrieve similar paintings. This search engine improved the painting retrieval performance.

For classifying paintings by artist, style or genre, only the color information from the paintings was used by the CNNs to perform classification; however, the spatial information [45] in images could also be used to improve classification tasks. In a given set of paintings, the artists, styles, genres and other features may be related. As we can clearly see, van Gogh's paintings have a high probability of being portrait and containing Post-Impressionism styles. Therefore, in future work, we will attempt to incorporate additional features to construct classification approaches based on both color features and other information. In the experiment, the models pretrained on ImageNet achieve the best effects for art classification; these results indicate that real-world image classification ability transfers well to the art world. In the future, we will use models pretrained on other datasets (similar to the large art datasets from OmniArt) [46], Art500K [47, 48], and similar databases from other fields, to determine which pretrained models perform best for art classification. We also plan to analyze how the pretrained parameters affect the classification results. Understanding how to extract features and classify them using computers is an important task. In this study, we used visualizations to detect which painting categories are easy for a computer to distinguish and which are more difficult. When looking at the paintings, people tend to focus on certain areas of interest [49]. Thus, in order to further explore the mechanism of computer understanding of paintings, saliency maps [50] could also be used in future research to investigate which areas the computer focuses on when classifying a painting.

## Author Contributions

**Conceptualization:** Wentao Zhao, Xinguo Qiu.

**Data curation:** Dalin Zhou.

**Formal analysis:** Wentao Zhao.

**Funding acquisition:** Xinguo Qiu, Wei Jiang.

**Investigation:** Wei Jiang.

**Supervision:** Xinguo Qiu.

**Validation:** Xinguo Qiu.

**Writing – review & editing:** Wentao Zhao, Dalin Zhou.

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
