## [Decision Letter · Decision Letter 0]

15 Jan 2021

PONE-D-20-34898

Compare the performance of the models in art classification

PLOS ONE

Dear Dr. Zhao,

Thank you for submitting your manuscript to PLOS ONE. After careful consideration, we feel that it has merit but does not fully meet PLOS ONE’s publication criteria as it currently stands. Therefore, we invite you to submit a revised version of the manuscript that addresses the points raised during the review process.

We look forward to receiving your revised manuscript.

Kind regards,

Khanh N.Q. Le

Academic Editor

PLOS ONE

Journal Requirements:

2. Please amend either the title on the online submission form (via Edit Submission) or the title in the manuscript so that they are identical.

3.We note that Figure(s) 1, 2, 3, 6 and 7 in your submission contain copyrighted images. All PLOS content is published under the Creative Commons Attribution License (CC BY 4.0), which means that the manuscript, images, and Supporting Information files will be freely available online, and any third party is permitted to access, download, copy, distribute, and use these materials in any way, even commercially, with proper attribution. For more information, see our copyright guidelines: http://journals.plos.org/plosone/s/licenses-and-copyright.

a)   You may seek permission from the original copyright holder of Figure(s) 1, 2, 3, 6 and 7 to publish the content specifically under the CC BY 4.0 license.

Reviewers' comments:

Reviewer's Responses to Questions

**Comments to the Author**

1. Is the manuscript technically sound, and do the data support the conclusions?

Reviewer #1: Partly

Reviewer #2: Yes

Reviewer #3: Yes

2. Has the statistical analysis been performed appropriately and rigorously? 

Reviewer #1: Yes

Reviewer #2: Yes

Reviewer #3: Yes

3. Have the authors made all data underlying the findings in their manuscript fully available?

Reviewer #1: Yes

Reviewer #2: Yes

Reviewer #3: Yes

4. Is the manuscript presented in an intelligible fashion and written in standard English?

Reviewer #1: Yes

Reviewer #2: Yes

Reviewer #3: Yes

5. Review Comments to the Author

Reviewer #1: This manuscript compared ResNet, ResNeXt, Res2Net, RegNetX, RegNetY and ResNeSt for art classification. The innovation of this work is limited. However, the comparison in this field is interesting. Some other problems in the manuscript are still concerned in the following:

1. The organization of this manuscript should be added to the end of the introduction.

2. The flow charts of the compared models are suggested to be added.

3. More details on the compared models should be exposed in the text.

4. The results could be analyzed in details.

Reviewer #2: After a careful review of this manuscript, I suggest my decision as accept after major revision. The following are some of my suggestions on this manuscript. I request the authors to make these required changes and resubmit the article after revisions.

The title of the article is stated as “Comparing the performances of deep learning models on art classification tasks”.

But I find no proper system architecture in the proposed contribution.

Adding an overall general architecture defining the entire system in the proposed section is much important.

Is there any particular reason to adopt DCNN algorithm?

What can be potential advantage from a performance perspective?

Need more clarity on feature extraction process.

The overall significance of the work is not well-defined

Results section needs much improvement.

The authors need to justify how the graphs defined in the result section contribute to the actual contribution of the work?

Adding more details on system configuration and types of tools used for simulation purpose along with appropriate specifications are more vital.

Add a real-world case study of the proposed scheme to understand the clarity of the work.

Organization of the paper require greater improvements.

Reviewer #3: 1- Overall, the key source of the article is unique and excellent. In the whole process, the outcome should be well considered and recommend for acceptable with significant corrections

2- There can be some resolutions problem with Fig.5 during publishing. If it is possible please re-draw them

3- Please comparing your work with other similar studies (related work)

4- The contribution and novelty should be better highlighted compared to the previous works

5- There is no mention of the source code oused by the authors in this study (especially the deep learning framework used). This prevents the reproducibility of the study.

6- “Please bring strong relevance to the scope of journal using most recent two years” literature further to improve readership. Plos One is always the key word. Importantly, please look at most recent published articles from species and gender identification of mosquito. Please compare the findings with relevant studies to over-repeat the results and draw some constructive conclusions.

7- An old deep learning model (ResNet), please implement a new architecture.

8- The models are selected mentioning they have a smaller number of layers. This should not be the case. The optimal models need to be selected for the datasets under study through architecture and hyperparameter optimization.

6. PLOS authors have the option to publish the peer review history of their article (what does this mean?). If published, this will include your full peer review and any attached files.

Reviewer #1: No

Reviewer #2: **Yes: **Dr. B. Santhosh Kumar

Reviewer #3: **Yes: **Mohamed Loey

---

## [Author Response · Author response to Decision Letter 0]

16 Feb 2021

We have responded to specific reviewer and editor comments in Response to Reviewers.pdf which have uploaded and labeled as Response to Reviewers. Thank you.

---

## [Decision Letter · Decision Letter 1]

26 Feb 2021

Compare the performance of the models in art classification

PONE-D-20-34898R1

Dear Dr. Zhao,

We’re pleased to inform you that your manuscript has been judged scientifically suitable for publication and will be formally accepted for publication once it meets all outstanding technical requirements.

Kind regards,

Khanh N.Q. Le

Academic Editor

PLOS ONE

Additional Editor Comments (optional):

Reviewers' comments:

Reviewer's Responses to Questions

**Comments to the Author**

1. If the authors have adequately addressed your comments raised in a previous round of review and you feel that this manuscript is now acceptable for publication, you may indicate that here to bypass the “Comments to the Author” section, enter your conflict of interest statement in the “Confidential to Editor” section, and submit your "Accept" recommendation.

Reviewer #1: All comments have been addressed

Reviewer #2: All comments have been addressed

2. Is the manuscript technically sound, and do the data support the conclusions?

Reviewer #1: Yes

Reviewer #2: Yes

3. Has the statistical analysis been performed appropriately and rigorously? 

Reviewer #1: Yes

Reviewer #2: Yes

4. Have the authors made all data underlying the findings in their manuscript fully available?

Reviewer #1: No

Reviewer #2: Yes

5. Is the manuscript presented in an intelligible fashion and written in standard English?

Reviewer #1: Yes

Reviewer #2: Yes

6. Review Comments to the Author

Reviewer #1: All my questions have been solved this time. In my opinion, the current version can be published directly.

Reviewer #2: The author addressed all the commands. So I strongly Recommended this article has been accepted for publication.

7. PLOS authors have the option to publish the peer review history of their article (what does this mean?). If published, this will include your full peer review and any attached files.

Reviewer #1: No

Reviewer #2: **Yes: **Dr B Santhosh Kumar

---

## [Editor Report · Acceptance letter]

3 Mar 2021

PONE-D-20-34898R1 

Compare the performance of the models in art classification 

Dear Dr. Qiu:

I'm pleased to inform you that your manuscript has been deemed suitable for publication in PLOS ONE. Congratulations! Your manuscript is now with our production department. 

Kind regards, 

on behalf of

Dr. Khanh N.Q. Le 

Academic Editor

PLOS ONE